# Minimax Optimal Fixed-Budget Best Arm Identification in Linear Bandits

**Junwen Yang**
Institute of Operations Research and Analytics
National University of Singapore
junwen_yang@u.nus.edu

**Vincent Y. F. Tan**
Department of Mathematics
Department of Electrical and Computer Engineering
Institute of Operations Research and Analytics
National University of Singapore
vtan@nus.edu.sg

## Abstract

We study the problem of best arm identification in linear bandits in the fixed-budget setting. By leveraging properties of the G-optimal design and incorporating it into the arm allocation rule, we design a parameter-free algorithm, Optimal Design-based Linear Best Arm Identification (OD-LinBAI). We provide a theoretical analysis of the failure probability of OD-LinBAI. Instead of all the optimality gaps, the performance of OD-LinBAI depends only on the gaps of the top $d$ arms, where $d$ is the effective dimension of the linear bandit instance. Complementarily, we present a minimax lower bound for this problem. The upper and lower bounds show that OD-LinBAI is minimax optimal up to constant multiplicative factors in the exponent, which is a significant theoretical improvement over existing methods (e.g., BayesGap, Peace, LinearExploration and GSE), and settles the question of ascertaining the difficulty of learning the best arm in the fixed-budget setting. Finally, numerical experiments demonstrate considerable empirical improvements over existing algorithms on a variety of real and synthetic datasets.

## 1 Introduction

The multi-armed bandit problem is a model that exemplifies the exploration-exploitation tradeoff in online decision making. It has various applications in drug design, online advertising, recommender systems, and so on. In stochastic multi-armed bandit problems, the agent sequentially chooses an arm from the given arm set at each time step and then observes a random reward drawn from the unknown distribution associated with the chosen arm.

The standard multi-armed bandit problem, where the arms are not correlated with one another, has been studied extensively in the literature. While the *regret minimization* problem aims at maximizing the cumulative rewards by the trade-off between exploration and exploitation [1–4], the *pure exploration* problem focuses on efficient exploration with specific goals, e.g., to identify the best arm [5–11]. There are two complementary settings for the problem of *best arm identification*: (i) Given $T \in \mathbb{N}$, the agent aims to maximize the probability of finding the best arm in at most $T$ time steps; (ii) Given $\delta > 0$, the agent aims to find the best arm with the probability of at least $1 - \delta$ in the smallest number of steps. These settings are respectively known as the fixed-budget and fixed-confidence settings.

36th Conference on Neural Information Processing Systems (NeurIPS 2022).

In this paper, we consider the problem of best arm identification in linear bandits in the fixed-budget setting. In linear bandits, the arms are correlated through an unknown global regression parameter vector $\theta^* \in \mathbb{R}^d$. In particular, each arm $i$ from the arm set $\mathcal{A}$ is associated with an arm vector $a(i) \in \mathbb{R}^d$, and the expected reward of arm $i$ is given by the inner product between $\theta^*$ and $a(i)$. Hence, the standard multi-armed bandits and linear bandits are fundamentally different due to the fact that for the latter, pulling one arm can indirectly reveal information about the other arms but in the former, the arms are independent.

A wide range of applications in practice can be modeled by linear bandits. For example, Tao et al. [12] considered online advertising, where the goal is to select an advertisement from a pool to maximize the probability of clicking for web users with different features. Empirically, the probability of clicking can be approximated by a linear combination of various attributes associated with the user and the advertisements (such as age, gender, the domain, keywords, advertising genres, etc.). Moreover, Hoffman et al. [13] applied the linear bandit model into the traffic sensor network problem and the problem of automatic model selection and algorithm configuration.

**Main contributions.** Our main contributions are as follows:

(i) We design an algorithm *Optimal Design-based Linear Best Arm Identification* (OD-LinBAI). This *computationally efficient* algorithm utilizes a phased elimination-based strategy in which the number of times each arm is pulled in each phase depends on G-optimal designs [14]. Besides, OD-LinBAI is *totally parameter-free*, whereas some existing methods (e.g., BayesGap and Peace) require the knowledge of the problem instance (which is typically not known in practice).

(ii) We derive an upper bound on the failure probability of OD-LinBAI. The failure probability is a significant improvement over those of existing methods which we survey in detail in Section 4.2. In particular, we show that the exponent of the failure probability depends on a hardness quantity $H_{2,\text{lin}}$. This quantity is a function of *only* the first $d - 1$ optimality gaps, where $d$ is the dimension of the arm vectors. This is a surprising and significant difference compared to the upper bounds of the failure probabilities of various algorithms for best arm identification in standard multi-armed bandits [7, 9, 15] and BayesGap [13] in linear bandits, which all depend on a hardness quantity that depends on *all* the gaps. Moreover, OD-LinBAI improves the exponent of the error probability by a factor of $\Theta(\log d)$ over Peace [16] in the worst-case sense or a factor of $\Theta((\log K)/(\log d))$ (which could be much larger than 1) over LinearExploration [17] and GSE [18] in general.

(iii) Lastly, using ideas from Carpentier and Locatelli [10], we prove a minimax lower bound which involves another hardness quantity $H_{1,\text{lin}}$. By comparing $H_{1,\text{lin}}$ to $H_{2,\text{lin}}$, we show that OD-LinBAI is minimax optimal up to constants in the exponent. OD-LinBAI is the first algorithm that provably achieves minimax optimality in this problem, and finally settles the question of ascertaining the hardness of learning the best arm in the fixed-budget setting for linear bandits. In addition, experiments in both synthetic and real-world datasets firmly corroborate the efficacy of OD-LinBAI vis-à-vis other existing methods.

**Related work.** The problem of regret minimization in linear bandits was first studied by Abe and Long [19], and has attracted extensive interest in the development of various algorithms (e.g., UCB-style algorithms [20–24], Thompson sampling [25, 26]). In particular, in the book of Lattimore and Szepesvári [27], a regret minimization algorithm based on the G-optimal design was proposed for linear bandits with finitely many arms. Although both this algorithm and our algorithm OD-LinBAI utilize the G-optimal design technique, they differ in numerous aspects including the manner of elimination and arm allocation, which emanates from the two different objectives.

For the problem of best arm identification in linear bandits, the fixed-confidence setting has previously been studied in [12, 28–34]. In particular, Soare et al. [28] introduced the optimal G-allocation problem and proposed a static algorithm $\mathcal{XY}$-Oracle as well as a semi-adaptive algorithm $\mathcal{XY}$-Adaptive; see Remark 2 for more discussions on Soare et al. [28]. Degenne et al. [32] treated the problem as a two-player zero-sum game between the agent and the nature, and thus designed an asymptotically optimal algorithm for the fixed-confidence setting.

The fixed-budget setting for the problem of best arm identification in linear bandits has also been studied in a few previous and concurrent works. Hoffman et al. [13] introduced a gap-based exploration algorithm BayesGap, which is a Bayesian treatment of UGapEb [8] for standard multi-armed

bandits. Peace by Katz-Samuels et al. [16] utilizes an experimental design based on the Gaussian-Width of the underlying arm set, which characterizes the geometry of the instance better in some instances. However, both BayesGap and Peace are computationally expensive and not parameter-free. Recently, Alieva et al. [17] introduced an elimination algorithm named LinearExploration, which is also robust to moderate levels of model misspecification. Generalized Successive Elimination (GSE) by Azizi et al. [18] shares a similar structure with LinearExploration and applies to generalized linear models. Nevertheless, none of the above is minimax optimal. See Section 4 and Section 5 for more comparisons between OD-LinBAI and other existing algorithms.

## 2  Problem setup and preliminaries

**Best arm identification in linear bandits.** We consider the standard linear bandit problem with an unknown global regression parameter. In a linear bandit instance $\nu$, the agent is given an arm set $\mathcal{A} = [K]$, which corresponds to known arm vectors $\{a(1), a(2), \ldots, a(K)\} \subset \mathbb{R}^d$. At each time $t$, the agent chooses an arm $A_t$ from the arm set $\mathcal{A}$ and then observes a noisy reward

$$X_t = \langle \theta^*, a(A_t) \rangle + \eta_t$$

where $\theta^* \in \mathbb{R}^d$ is the unknown parameter vector and $\eta_t$ is independent zero-mean 1-subgaussian random noise.

In the fixed-budget setting, given a time budget $T \in \mathbb{N}$, the agent aims at maximizing the probability of identifying the best arm, i.e., the arm with the largest expected reward, with no more than $T$ arm pulls. More formally, the agent uses an *online* algorithm $\Pi$ to decide the arm $A_t^\Pi$ to pull at each time step $t$, and the arm $i_{\text{out}}^\Pi \in \mathcal{A}$ to output as the identified best arm by time $T$. We abbreviate $A_t^\Pi$ as $A_t$ and $i_{\text{out}}^\Pi$ as $i_{\text{out}}$ when there is no ambiguity.

For any arm $i \in \mathcal{A}$, let $p(i) = \langle \theta^*, a(i) \rangle$ denote the expected reward. For convenience, we assume that the expected rewards of the arms are in descending order and the best arm is unique. That is to say, $p(1) > p(2) \geq \cdots \geq p(K)$. For any suboptimal arm $i$, we denote $\Delta_i = p(1) - p(i)$ as the optimality gap. For ease of notation, we also set $\Delta_1 = \Delta_2$. Furthermore, let $\mathcal{E}$ denote the set of all the linear bandit instances defined above.

**Dimensionality-reduced arm vectors.** For any linear bandit instance, if the corresponding arm vectors do not span $\mathbb{R}^d$, i.e., $\text{span}(\{a(1), a(2), \ldots, a(K)\}) \subsetneq \mathbb{R}^d$, the agent can work with a set of dimensionality-reduced arm vectors $\{a'(1), a'(2), \ldots, a'(K)\} \subset \mathbb{R}^{d'}$, that spans $\mathbb{R}^{d'}$, with little consequence. Specifically, let $B \in \mathbb{R}^{d \times d'}$ be a matrix whose columns form an orthonormal basis of the subspace spanned by $a(1), a(2), \ldots, a(K)$.[1] Then the agent can simply set $a'(i) = B^\top a(i)$ for each arm $i$. To verify this, notice that $BB^\top$ is a projection matrix onto the subspace spanned by $\{a(1), a(2), \ldots, a(K)\}$ and consequently

$$p(i) = \langle \theta^*, a(i) \rangle = \langle \theta^*, BB^\top a(i) \rangle = \langle B^\top \theta^*, B^\top a(i) \rangle = \langle \theta^{*\prime}, a'(i) \rangle .$$

Note that $\theta^*$ is the unknown parameter vector for original arm vectors while $\theta^{*\prime} = B^\top \theta^*$ is the corresponding unknown parameter vector for the dimensionality-reduced arm vectors. In the problem of linear bandits, what we really care about is not the original unknown parameter $\theta^*$ itself but the inner products between $\theta^*$ and the arm vectors $a(i)$, which establishes the equivalence of original arm vectors and dimensionality-reduced arm vectors.

In our work, without loss of generality, we assume that the entire set of original arm vectors $\{a(1), a(2), \ldots, a(K)\}$ span $\mathbb{R}^d$ and $d \geq 2$.[2] However, this idea of transforming into dimensionality-reduced arm vectors is often used in our elimination-based algorithm. See Section 3 for details.

**Least squares estimators.** Let $A_1, A_2, \ldots, A_n$ be the sequence of arms pulled by the agent and $X_1, X_2, \ldots, X_n$ be the corresponding noisy rewards. Suppose that the corresponding arm vectors $\{a(A_1), a(A_2), \ldots, a(A_n)\}$ span $\mathbb{R}^d$, then the ordinary least squares (OLS) estimator of $\theta^*$ is given by

$$\hat{\theta} = V^{-1} \sum_{t=1}^{n} a(A_t) X_t$$

---

[1] Such an orthonormal basis can be calculated efficiently with the reduced singular value decomposition, Gram–Schmidt process, etc.

[2] The situation that $d = 1$ is trivial: each arm vector is a scalar multiple of one another.

where $V = \sum_{t=1}^{n} a(A_t)a(A_t)^\top \in \mathbb{R}^{d \times d}$ is invertible. By applying the properties of subgaussian random variables, a confidence bound for the OLS estimator can be derived as follows.

**Proposition 1** (Lattimore and Szepesvári [27, Chapter 20]). *If $A_1, A_2, \ldots, A_n$ are deterministically chosen without knowing the realizations of $X_1, X_2, \ldots, X_n$, then for any $a \in \mathbb{R}^d$ and $\delta > 0$,*

$$\Pr\left[\langle \hat{\theta} - \theta^*, a \rangle \geq \sqrt{2\|a\|_{V^{-1}}^2 \log\left(\frac{1}{\delta}\right)}\right] \leq \delta.$$

*Remark* 1. When the arm pulls are adaptively chosen according to the random rewards, Proposition 1 no longer applies and an extra factor $\sqrt{d}$ has to be paid for adaptive arm pulls [23]. Our algorithm avoids this issue by deciding the arm pulls at the beginning of each phase, and designing the OLS estimator only based on the information from the current phase. See Section 3 for details.

**G-optimal design.** The confidence interval in Proposition 1 shows the strong connection between the arm allocation in linear bandits and experimental design theory [35]. To control the confidence bounds, we first introduce the G-optimal design technique into the problem of best arm identification in linear bandits in the fixed-budget setting. Formally, the G-optimal design problem aims at finding a probability distribution $\pi : \{a(i) : i \in \mathcal{A}\} \to [0, 1]$ that minimises

$$g(\pi) = \max_{i \in \mathcal{A}} \|a(i)\|_{V(\pi)^{-1}}^2$$

where $V(\pi) = \sum_{i \in \mathcal{A}} \pi(a(i))a(i)a(i)^\top$. Theorem 1 states the existence of a small-support G-optimal design and the minimum value of $g$.

**Theorem 1** (Kiefer and Wolfowitz [14]). *If the arm vectors $\{a(i) : i \in \mathcal{A}\}$ span $\mathbb{R}^d$, the following statements are equivalent: (i) $\pi^*$ is a minimiser of $g$; (ii) $\pi^*$ is a maximiser of $f(\pi) = \log \det V(\pi)$; (iii) $g(\pi^*) = d$. Furthermore, there exists a minimiser $\pi^*$ of $g$ such that $|\operatorname{Supp}(\pi^*)| \leq d(d+1)/2$.*

*Remark* 2. It is worth mentioning that the G-optimal design problem for finite arm vectors is a convex optimization problem while the original G-allocation problem in Soare et al. [28] for the fixed-confidence best arm identification in linear bandits is an NP-hard discrete optimization problem. A classical algorithm to solve the G-optimal design problem is the Frank–Wolfe algorithm [36], whose modified version guarantees linear convergence [37]. For our work, it is sufficient to compute an $\epsilon$-approximate optimal design[3] with minimal impact on performance. Recently, a near-optimal design with smaller support was proposed in Lattimore et al. [38], which might be helpful in some scenarios. See Appendix A for more discussions on the above issues. To reduce clutter and ease the reading, henceforward in the main text, we assume that a G-optimal design for finite arm vectors can be found accurately and efficiently.

## 3 Algorithm

Pseudocode for our algorithm *Optimal Design-based Linear Best Arm Identification* (OD-LinBAI) is presented in Algorithm 1.

The algorithm partitions the whole horizon into $\lceil \log_2 d \rceil$ phases, and maintains an *active* arm set $\mathcal{A}_r$ in each phase $r$. The length of each phase roughly equals $m$, which will be formally defined in (1).

Motivated by the equivalence of the original arm vectors and the dimensionality-reduced arm vectors, at the beginning of each phase $r$, the algorithm computes a set of dimensionality-reduced arm vectors $\{a_r(i) : i \in \mathcal{A}_{r-1}\} \subset \mathbb{R}^{d_r}$ which spans the $d_r$-dimensional Euclidean space $\mathbb{R}^{d_r}$. This can be implemented based on the dimensionality-reduced arm vectors of the last phase $\{a_{r-1}(i) : i \in \mathcal{A}_{r-1}\}$ in an iterative manner (Lines $5 - 11$).

After that, Algorithm 1 finds a G-optimal design $\pi_r$ for the current dimensionality-reduced arm vectors, with a restriction on the cardinality of the support when $r = 1$. OD-LinBAI then pulls each arm in $\mathcal{A}_{r-1}$ according to the proportions specified by the optimal design $\pi_r$. Specifically, the algorithm chooses each arm $i \in \mathcal{A}_{r-1}$ exactly $T_r(i) = \lceil \pi_r(a_r(i)) \cdot m \rceil$ times, where the parameter $m$ is fixed among different phases and defined as

$$m = \frac{T - \min(K, \frac{d(d+1)}{2}) - \sum_{r=1}^{\lceil \log_2 d \rceil - 1} \lceil \frac{d}{2^r} \rceil}{\lceil \log_2 d \rceil}. \tag{1}$$

---

[3]For an $\epsilon$-approximate optimal design $\pi$, $g(\pi) \leq (1 + \epsilon)d$.

**Algorithm 1** Optimal Design-based Linear Best Arm Identification (OD-LinBAI)

---

**Input:** time budget $T$, arm set $\mathcal{A} = [K]$ and arm vectors $\{a(1), a(2), \ldots, a(K)\} \subset \mathbb{R}^d$.

1: Initialize $t_0 = 1$, $\mathcal{A}_0 \leftarrow \mathcal{A}$ and $d_0 = d$.
2: For each arm $i \in \mathcal{A}_0$, set $a_0(i) = a(i)$.
3: Calculate $m$ using Equation (1).
4: **for** $r = 1$ to $\lceil \log_2 d \rceil$ **do**
5:      Set $d_r = \dim\left(\text{span}\left(\{a_{r-1}(i) : i \in \mathcal{A}_{r-1}\}\right)\right)$.
6:      **if** $d_r = d_{r-1}$ **then**
7:          For each arm $i \in \mathcal{A}_{r-1}$, set $a_r(i) = a_{r-1}(i)$.
8:      **else**
9:          Find matrix $B_r \in \mathbb{R}^{d_{r-1} \times d_r}$ whose columns form a orthonormal basis of the subspace spanned by $\{a_{r-1}(i) : i \in \mathcal{A}_{r-1}\}$.
10:         For each arm $i \in \mathcal{A}_{r-1}$, set $a_r(i) = B_r^\top a_{r-1}(i)$.
11:      **end if**
12:      **if** $r = 1$ **then**
13:          Find a G-optimal design $\pi_r : \{a_r(i) : i \in \mathcal{A}_{r-1}\} \to [0, 1]$ with $|\text{Supp}(\pi_r)| \leq \frac{d(d+1)}{2}$.
14:      **else**
15:          Find a G-optimal design $\pi_r : \{a_r(i) : i \in \mathcal{A}_{r-1}\} \to [0, 1]$.
16:      **end if**
17:      Set

$$T_r(i) = \lceil \pi_r(a_r(i)) \cdot m \rceil \quad \text{and} \quad T_r = \sum_{i \in \mathcal{A}_{r-1}} T_r(i).$$

18:      Choose each arm $i \in \mathcal{A}_{r-1}$ exactly $T_r(i)$ times.
19:      Calculate the OLS estimator:

$$\hat{\theta}_r = V_r^{-1} \sum_{t=t_r}^{t_r + T_r - 1} a_r(A_t) X_t \quad \text{with} \quad V_r = \sum_{i \in \mathcal{A}_{r-1}} T_r(i) a_r(i) a_r(i)^\top.$$

20:      For each arm $i \in \mathcal{A}_{r-1}$, estimate the expected reward:

$$\hat{p}_r(i) = \langle \hat{\theta}_r, a_r(i) \rangle.$$

21:      Let $\mathcal{A}_r$ be the set of $\lceil d/2^r \rceil$ arms in $\mathcal{A}_{r-1}$ with the largest estimates of the expected rewards.
22:      Set $t_{r+1} = t_r + T_r$.
23: **end for**

**Output:** the only arm $i_{\text{out}}$ in $\mathcal{A}_{\lceil \log_2 d \rceil}$.

---

Note that $m = \Theta(T / \log_2 d)$ as $T \to \infty$ with $K$ fixed. Lemma 1 in Appendix B shows with such choice of $m$, the total time budget consumed by the agent is no more than $T$. The parameter $m$ plays a significant role in the implementation as well as the theoretical analysis of Algorithm 1.

Since the support of the G-optimal design $\pi_r$ must span $\mathbb{R}^{d_r}$, the OLS estimator can be directly applied (Line 19). Then for each arm $i \in \mathcal{A}_{r-1}$, an estimate of the expected reward is derived. Algorithm 1 decouples the estimates of different phases and only utilizes the information obtained in the current phase $r$.

At the end of each phase $r$, Algorithm 1 eliminates a subset of possibly suboptimal arms. In particular, $K - \lceil d/2 \rceil$ arms are eliminated in the first phase and about half of the active arms are eliminated in each of the following phases. Eventually, there is only single arm $i_{\text{out}}$ in the active set, which is the output of Algorithm 1.

*Remark* 3. It is worth considering the case of standard multi-armed bandits, which can be modeled as a special case of linear bandits. In particular, for any arm $i \in \mathcal{A} = [K]$, the corresponding arm vector is chosen to be $e_i$, which is the $i^{\text{th}}$ standard basis vector of $\mathbb{R}^K$. It follows that $d = K$, $\theta^* = [p(1), p(2), \ldots, p(K)]^\top \in \mathbb{R}^K$ and arms are not correlated with one another. A simple mathematical derivation shows that we can always use a set of standard basis vectors of $\mathbb{R}^{d_r}$ to represent the arm vectors regardless of which arms remain active during phase $r$. Also, the G-optimal design for a set of standard basis vectors is the uniform distribution on all of the active arms. Since

pulling one arm does not provide information about the other arms, the empirical estimates based on the OLS estimator are exactly the empirical means. Altogether, for standard multi-armed bandits, OD-LinBAI reduces to the procedure of Sequential Halving [9], which is a state-of-the-art algorithm for best arm identification in standard multi-armed bandits in the fixed-budget setting.

*Remark* 4. The G-optimal design steps in Lines 13 and 15 in OD-LinBAI may be replaced by the $\mathcal{XY}$-allocation [28] or other techniques in experimental designs. However, our work focuses on establishing minimax optimality and thus the application of G-optimal designs, which optimize over the worst cases, is natural. The $\mathcal{XY}$-allocation may result in better empirical performance but the improvement might be limited or even absent in worst-case scenarios. More importantly, as noted in Degenne et al. [32, Remark 1], for the general $\mathcal{XY}$-allocation problem, only heuristic solutions can be obtained (without convergence guarantees). Nevertheless, the G-optimal design problem can be provably solved with a linear convergence guarantee [37]. Overall, the implementation of OD-LinBAI is computationally very efficient.

## 4 Main results

### 4.1 Upper bound

We first state an upper bound on the error probability of OD-LinBAI (Algorithm 1). The proof of Theorem 2 is deferred to Appendix B.

**Theorem 2.** *For any linear bandit instance $\nu \in \mathcal{E}$, OD-LinBAI outputs an arm $i_{\text{out}}$ satisfying*

$$\Pr\left[i_{\text{out}} \neq 1\right] \leq \left(\frac{4K}{d} + 3\log_2 d\right) \exp\left(-\frac{m}{32H_{2,\text{lin}}}\right)$$

*where $m$ is defined in Equation (1) and*

$$H_{2,\text{lin}} = \max_{2 \leq i \leq d} \frac{i}{\Delta_i^2}.$$

Theorem 2 shows the error probability of OD-LinBAI is upper bounded by

$$\exp\left(-\Omega\left(\frac{T}{H_{2,\text{lin}}\log_2 d}\right)\right) \tag{2}$$

which depends on $T$, $d$ and $H_{2,\text{lin}}$. We remark that none of the three terms is avoidable in view of our lower bounds (see Section 4.3).

In particular, $T$ is the time budget of the problem and $d$ is the effective dimension of the arm vectors.[4] Given $T$ and $d$, $H_{2,\text{lin}}$ quantifies the difficulty of identifying the best arm in the linear bandit instance. The parameter $H_{2,\text{lin}}$ generalizes its analogue

$$H_2 = \max_{2 \leq i \leq K} \frac{i}{\Delta_i^2}$$

proposed by Audibert et al. [7] for standard multi-armed bandits. However, $H_{2,\text{lin}}$ is not larger than $H_2$ since $H_{2,\text{lin}}$ is only a function of the first $d-1$ optimality gaps while $H_2$ considers all of the $K-1$ optimality gaps. In the extreme case that all of the suboptimal arms have the same optimality gaps, i.e., $\Delta_2 = \Delta_3 = \cdots = \Delta_K$, the two terms $H_2$ and $H_{2,\text{lin}}$ can differ significantly. In general, we have

$$H_{2,\text{lin}} \leq H_2 \leq \frac{K}{d}H_{2,\text{lin}}$$

and both inequalities are essentially sharp, i.e., can be achieved by some linear bandit instances. This highlights a major difference between best arm identification in the fixed-budget setting for linear bandits and standard multi-armed bandits. Due to the linear structure, arms are correlated and we can estimate the mean reward of one arm with the help of the other arms. Thus, the hardness quantity $H_{2,\text{lin}}$ is only a function of the top $d$ arms rather than all the arms.

---

[4]Recall that we assume the entire set of original arm vectors $\{a(1), a(2), \ldots, a(K)\}$ span $\mathbb{R}^d$.

## 4.2 Comparisons to other algorithms

We compare OD-LinBAI and other existing algorithms with respect to the algorithm design as well as the theoretical guarantees in the following.

**Comparisons to BayesGap [13].**

(i) The model used in BayesGap [13] is based on Bayesian linear bandits, where the unknown parameter vector $\theta^*$ is drawn from a known prior distribution $\mathcal{N}(0, \eta^2 I)$ and the additive noise is required to be Gaussian. However, OD-LinBAI does not require these assumptions and the upper bound holds for any deterministic or random $\theta^* \in \mathbb{R}^d$.

(ii) The algorithm and theoretical guarantee of BayesGap explicitly require the knowledge of a hardness quantity $H_1 = \sum_{1 \leq i \leq K} \Delta_i^{-2}$ to control the confidence region and then allocate exploration. However, this hardness quantity $H_1$ is almost always unknown to the agent in practice. In most practical applications, BayesGap has to estimate $H_1$ in an adaptive way, which works reasonably well in numerical experiments but lacks theoretical guarantees.

(iii) BayesGap's error probability is upper bounded by

$$\exp\left(-\Omega\left(\frac{T}{H_1}\right)\right) \tag{3}$$

which depends on $T$ and $H_1$. Compared with (3), the upper bound of OD-LinBAI in (2) has an extra $\log_2 d$ term. This is an interesting phenomena which is also present in standard multi-armed bandits [7, 10]. For best arm identification in standard multi-armed bandits, without the knowledge of the hardness quantity $H_1$, the agent has to pay a price of $\log_2 K$ for the adaptation to the problem complexity. In Theorem 3, we prove a similar result for linear bandits, in which the price of adaptation is $\log_2 d$.

The upper bound (3) involves $H_1$, a function of *all* the optimality gaps. It holds that $H_1 \geq H_2 \geq H_{2,\text{lin}}$. Thus, the upper bound of OD-LinBAI is not worse (and often better) in its dependence on the hardness/complexity parameter.

**Comparisons to Peace [16] (Also see Appendix C).**

(i) To ensure there is only a *single* arm in the final active set, the fixed-budget version of Peace requires $\gamma(\{a(i), a(1)\}) \geq 1$ for all suboptimal arms $i \neq 1$ (where $\gamma(\cdot)$ is defined in Katz-Samuels et al. [16]). Note that this is not only a requirement for the theoretical bound but also a requirement for the *feasibility* of the algorithm. If this inequality is not satisfied, the linear bandit instance needs to be "rescaled" before the algorithm is run, resulting in a larger bound on the error probability. In practice, the best arm is unknown and the rescaling factor can thus only be conservatively bounded as $\min_i \gamma(\{a(i), a(1)\}) \geq \min_{i,j} \gamma(\{a(i), a(j)\})$. However, the latter quantity can be miniscule. In particular, if there exist two arms that are nearly identical, i.e., $\min_{i,j} \gamma(\{a(i), a(j)\})$ is very small, the bound on the error probability may be larger than 1, and hence vacuous. Besides, the algorithm may terminate with most of its time budget wasted. In contrast, OD-LinBAI is *fully parameter-free* and does not require any information about the instance.

(ii) It is not straightforward to compare the error probabilities of OD-LinBAI and Peace in general since Peace involves some tricky terms that do not admit closed-form expressions. Here we consider the special case of standard multi-armed bandits (as discussed in Remark 3) with all optimality gaps equal to the minimal one $\Delta_1$. In this case $\rho^* = \Theta(\Delta_1^{-2} \cdot d)$, $\gamma^* = \Theta(\Delta_1^{-2} \cdot d \log d)$ and $\log(\gamma(\mathcal{Z})) = \Theta(\log d)$; these terms appear in the denominator of the exponent in Peace's bound on the error probability. Therefore, the error probability of Peace is $\exp\left(-\Omega\left(\frac{T\Delta_1^2}{d \log^2 d}\right)\right)$ while ours is $\exp\left(-\Omega\left(\frac{T\Delta_1^2}{d \log d}\right)\right)$, which also shows Peace is *not* minimax optimal in the exponent in view of our lower bounds, to be presented in Section 4.3. See Appendix C for the precise details of the above derivations.

**Comparisons to LinearExploration [17] and GSE [18].**

(i) The idea of elimination has been well-received and is ubiquitous in linear bandits. Although LinearExploration [17], GSE [18] and OD-LinBAI all leverage this idea, we emphasize that the elimination criteria for these algorithms are different. In particular, OD-LinBAI divides the time budget into roughly $\log_2 d$ phases while the other algorithms divide the budget into roughly $\log_2 K$ phases. Additionally, OD-LinBAI always controls the *dimension* of the active set in each phase, using the dimensionality reduction techinique in Section 2.

Table 1: Comparisons of different hardness quantities: $H_1$, $H_2$, $H_{1,\mathrm{lin}}$ and $H_{2,\mathrm{lin}}$.

| $H_1 = \sum_{1 \leq i \leq K} \Delta_i^{-2}$ | $H_2 = \max_{2 \leq i \leq K} i \cdot \Delta_i^{-2}$ | $1 \leq H_1/H_2 \leq \log(2K)$ [7] |
|---|---|---|
| $H_{1,\mathrm{lin}} = \sum_{1 \leq i \leq d} \Delta_i^{-2}$ | $H_{2,\mathrm{lin}} = \max_{2 \leq i \leq d} i \cdot \Delta_i^{-2}$ | $1 \leq H_{1,\mathrm{lin}}/H_{2,\mathrm{lin}} \leq \log(2d)$ |
| $1 \leq H_1/H_{1,\mathrm{lin}} \leq K/d$ | $1 \leq H_2/H_{2,\mathrm{lin}} \leq K/d$ | |

(ii) The error probabilities of LinearExploration and GSE are upper bounded by $\exp\big(-\Omega(\frac{T}{\tilde{H}_2 \log_2 K})\big)$ and $\exp\big(-\Omega(\frac{T\Delta_1^2}{d \log_2 K})\big)$ respectively. Note that $K \geq d$, $H_{2,\mathrm{lin}} \leq d/\Delta_1^2$, and the hardness quantity $\tilde{H}_2$ in Alieva et al. [17] is of the same order as $H_{2,\mathrm{lin}}$. Hence, our exponent of the bound on the error probability is an improvement over their exponents by a factor of $\Theta((\log_2 K)/(\log_2 d))$, which may be much larger than 1.

## 4.3 Lower bound

Before stating the lower bound formally, we introduce

$$H_{1,\mathrm{lin}} = \sum_{1 \leq i \leq d} \Delta_i^{-2}.$$

This quantity is a generalization of $H_1$ that characterizes the difficulty of a linear bandit instance. This parameter is also associated with the top $d$ arms similarly to $H_{2,\mathrm{lin}}$. See Table 1 for a thorough comparison on different hardness quantities.

For any linear bandit instance $\nu \in \mathcal{E}$, we denote the hardness quantity $H_{1,\mathrm{lin}}$ of $\nu$ as $H_{1,\mathrm{lin}}(\nu)$.[5] In addition, let $\mathcal{E}(h)$ denote the set of linear bandit instances in $\mathcal{E}$ whose hardness parameter $H_{1,\mathrm{lin}}$ is upper bounded by $h$ (for some $h > 0$), i.e., $\mathcal{E}(h) = \{\nu \in \mathcal{E} : H_{1,\mathrm{lin}}(\nu) \leq h\}$.

**Theorem 3.** *If $T \geq h^2 \log(6Td)/900$, then*

$$\min_{\Pi} \max_{\nu \in \mathcal{E}(h)} \Pr\left[i_{\mathrm{out}}^{\Pi} \neq 1\right] \geq \frac{1}{6} \exp\left(-\frac{240T}{h}\right).$$

*Further if $h \geq 15d^2$, then*

$$\min_{\Pi} \max_{\nu \in \mathcal{E}(h)} \left(\Pr\left[i_{\mathrm{out}}^{\Pi} \neq 1\right] \cdot \exp\left(\frac{2700T}{H_{1,\mathrm{lin}}(\nu) \log_2 d}\right)\right) \geq \frac{1}{6}.$$

The proof of Theorem 3 is deferred to Appendix D. We emphasize that even though the proof of the lower bound follows some common ideas behind the proofs of most minimax lower bounds in bandit algorithms for various purposes, its value does not lie in its technical novelty, but rather that the result is *tight* vis-à-vis the upper bound we have derived based on the OD-LinBAI algorithm. The usual strategy, which is the strategy we adopt here, is to construct and analyze specific hard instances. In particular, we leverage the instances in Carpentier and Locatelli [10] for standard multi-armed bandits to construct hard linear bandit instances for any arbitrary $K$ and $d$. We discuss the tightness of the lower bound in the following.

Theorem 3 first shows that for any best arm identification algorithm $\Pi$, even with the knowledge of an upper bound $h$ on the hardness quantity $H_{1,\mathrm{lin}}$, there exists a linear bandit instance such that the error probability is at least

$$\exp\left(-O\left(\frac{T}{h}\right)\right). \tag{4}$$

Furthermore, for any best arm identification algorithm $\Pi$, without the knowledge of an upper bound $h$ on the hardness quantity $H_{1,\mathrm{lin}}$, there exists a linear bandit instance $\nu$ such that the error probability is at least

$$\exp\left(-O\left(\frac{T}{H_{1,\mathrm{lin}}(\nu) \log_2 d}\right)\right). \tag{5}$$

---

[5]When there is no ambiguity, $H_{1,\mathrm{lin}}$ will also be used.

Comparing the lower bounds (4) and (5) in two different settings, we show that the agent has to pay a price of $\log_2 d$ in the absence of the knowledge about the problem complexity. Finding a best arm identification algorithm that matches the lower bound (4) remains an open problem since the upper bound of BayesGap (3) involves $H_1$ but not $H_{1,\text{lin}}$. However, notice that the knowledge about the complexity quantity which is required for BayesGap is usually unavailable in real-life applications.

Now we compare the upper bound on the error probability of OD-LinBAI in (2) with the lower bound (5). Table 1 shows that $H_{1,\text{lin}} \geq H_{2,\text{lin}}$ always holds. Therefore, the upper bound in (2) is not larger than the lower bound in (5) in the exponent up to absolute constants. This shows OD-LinBAI (Algorithm 1) is *minimax optimal* up to multiplicative factors in the exponent and the upper bound cannot be improved in an order-wise sense in the exponent in general. At the same time, note that the upper bound holds for *all* instances while the lower bound is a minimax result which holds for *specific* instances. Since an upper bound can never be smaller than a lower bound, we know that the difficult instances for the problem of best arm identification in linear bandits in the fixed-budget setting are those whose $H_{1,\text{lin}}$ and $H_{2,\text{lin}}$ are of the same order.

## 5 Numerical experiments

In this section, we evaluate the performance of our algorithm OD-LinBAI and compare it with Sequential Halving [9], BayesGap [13], Peace [16], LinearExploration [17] and GSE [18]. For BayesGap, there are two versions: one is BayesGap-Oracle, which is given the exact information of the required hardness quantity $H_1$; the other is BayesGap-Adaptive, which adaptively estimates the hardness quantity by the three-sigma rule. In each setting, the reported error probabilities of different algorithms are averaged over $1024$ independent trials and the (tiny) error bars indicate the standard errors of the error probabilities. We present the results of one synthetic dataset here. Additional implementation details and numerical results (including another synthetic dataset, one real-world dataset and comparison to the recent LT&S algorithm for best arm identification in linear bandits with fixed confidence [33]) are provided in Appendix E.

### 5.1 Synthetic dataset 1: a hard instance

This benchmark dataset, in which there are numerous competitors for the second best arm, was considered for the problem of best arm identification in linear bandits in the fixed-confidence setting [30, 31, 33]. Similarly, we consider the situation that $d = 2$ and $K \geq 3$. We assume that the additive random noise follows the standard Gaussian distribution $\mathcal{N}(0,1)$. For simplicity, we set the unknown parameter vector $\theta^* = [1,0]^\top$. There is one best arm and one worst arm, which correspond to the arm vectors $a(1) = [1,0]^\top$ and $a(K) = [\cos(3\pi/4), \sin(3\pi/4)]^\top$ respectively. For any arm $i \in \{2,3,\ldots,K-1\}$, the corresponding arm vector is chosen to be $a(i) = [\cos(\pi/4 + \phi_i), \sin(\pi/4 + \phi_i)]^\top$ with $\phi_i$ drawn independently from $\mathcal{N}(0, 0.09^2)$. Therefore, there are $K - 2$ almost second best arms. Considering the definitions of four hardness quantities, it holds that $H_1 \approx H_2 \approx \frac{K}{d} H_{1,\text{lin}} \approx \frac{K}{d} H_{2,\text{lin}}$. Hence this is a hard instance in the sense that the linear structure is ex-

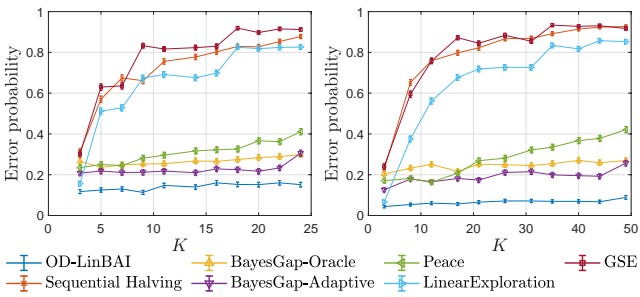

Figure 1: Error probabilities for different numbers of arms $K$ with $T = 25, 50$ from left to right.

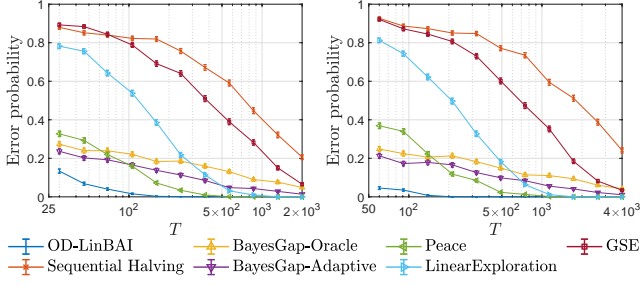

Figure 2: Error probabilities for different time budgets $T$ with $K = 25, 50$ from left to right.

tremely strong. A good algorithm needs to fully utilize the correlations of the arms to obtain information as efficiently as possible.

The experimental results with fixed $T$ and $K$ are presented in Figure 1 and Figure 2 respectively. In terms of this hard linear bandit instance, OD-LinBAI is clearly superior compared to its competitors. In fact, OD-LinBAI consistently pulls only one arm from the $K - 2$ almost second best arms and thus suffers minimal impact from the increase in $K$.

## 6 Conclusions and Future Work

We introduce the G-optimal design technique into the problem of best arm identification in linear bandits in the fixed-budget setting. We design a parameter-free and efficient algorithm OD-LinBAI. To characterize the difficulty of a linear bandit instance, we introduce two hardness quantities $H_{1,\mathrm{lin}}$ and $H_{2,\mathrm{lin}}$. The upper bound of the error probability of OD-LinBAI and the minimax lower bound of this problem are respectively characterized by $H_{1,\mathrm{lin}}$ and $H_{2,\mathrm{lin}}$ instead of their analogues $H_1$ and $H_2$ in standard multi-armed bandits. For the first time, minimax optimality (up to constant multiplicative factors in the exponent) has been achieved in this problem. While we submit that the ingredients that constitute OD-LinBAI are not surprising in the bandit literature, an open problem thus far has hence been solved in this contribution (by the careful derivation of an upper bound on the error probability of OD-LinBAI and an accompanying minimax lower bound). Our theoretical findings are also supported by the considerable improvements of the empirical performance of OD-LinBAI vis-à-vis existing algorithms on benchmark datasets.

A direction for future work is to design an instance-dependent asymptotically optimal algorithm for this problem. However, finding such an algorithm or an instance-dependent asymptotic lower bound for the problem of best arm identification in standard (i.e., $K$-armed) multi-armed bandits in the fixed-budget setting remains open. Finally, as Thompson sampling [1, 4] has been successfully extended to pure exploration in standard multi-armed bandits [39–42], it is interesting to study whether this technique can be generalized to *linear* bandits, in both the fixed-budget and fixed-confidence settings.

## Acknowledgments and Disclosure of Funding

This research/project is supported by the National Research Foundation Singapore and DSO National Laboratories under the AI Singapore Programme (AISG Award No: AISG2-RP-2020-018) and by Singapore Ministry of Education (MOE) AcRF Tier 1 Grants (A-0009042-01-00 and A-8000189-01-00).

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
