# OpenReview forum: "Minimax Optimal Fixed-Budget Best Arm Identification in Linear Bandits"
_NeurIPS.cc/2022/Conference — NeurIPS 2022 Accept_

### Official Review · Reviewer_RdL2 · 2022-07-10

**Rating:** 5
**Confidence:** 4
**Soundness:** 2 fair
**Presentation:** 4 excellent
**Contribution:** 2 fair

**Summary:**

This work studies linear bandit with the goal of identifying the best arm. The goal is to minimize the error probability when the best arm is not identified with limited number of pulls. The contribution of this paper is three-fold: 1) The author(s) introduced an elimination-based algorithm called OD-LinBAI utilizing the G-optimal design. Compared with the prior arts, the proposed one is parameter-free. 2) The author(s) also presented a lower bound, which shows that OD-LinBAI is nearly optimal. 3) Experiment results are presented to demonstrate the superior performance of the proposed algorithm.

**Questions:**

When the author(s) discuss the optimality of the proposed algorithm, $\Omega( \frac{K}{d} + \log_2(d) )$ is omitted. Could the author(s) help clarify the reason why such parameter can be omitted? Under the current presentation, it is hard for me to believe the proposed algorithm is minimax optimal.

**Strengths And Weaknesses:**

* Strengths

** This paper is written well and in good presentation which make it easy to follow.
** The newly proposed algorithm OD-LinBAI is parameter-free and computation-efficient, which makes it easier to be applied to real-world applicaitons.
** Both synthetic and real-world experiments are provided to prove the superior performance of the proposed policy.

* Weaknesses

** The ideas and techniques applied in this paper are not new. Specifically, the proposed algorithm is quite similar to the one proposed in [1] although the latter one targets best arm identification under fixed-confidence setting. I am not saying it is a bad thing since the goals of the problems in these two papers are different. However this will harm the originality of this paper.

[1] Chao Tao, Saúl Blanco, and Yuan Zhou. Best arm identification in linear bandits with linear
403 dimension dependency. In International Conference on Machine Learning, pages 4877–4886.
404 PMLR, 2018.

** There is still some gap between the upper and lower bounds despite the constant parameters inside the exponent. In particular, the parameter is $\Omega( \frac{K}{d} + \log_2(d) )$.

---

> ### Author Response · Authors · 2022-07-29
> **Response to Reviewer RdL2**
>
> We thank the reviewer for the meticulous reading and constructive comments.
> We have carefully addressed your concerns in the following.
>
> - The idea of elimination is  ubiquitous in both standard multi-armed bandits and linear bandits. The key challenge is how to **divide the time horizon into phases** and **choose the right criterion of elimination**. In particular, in our work, the lengths of the roughly $\log_2 d$ phases are almost equal and a fixed number of arms are eliminated altogether at the end of each phase, which is uncommon in the fixed-confidence setting. Our algorithm is somewhat similar to the one in [1] due to the inherent connection between the fixed-budget setting and the fixed-confidence setting. However, this connection is fairly tenuous, as has been well documented in [2-4].
>
>
> - Similarly to most works on best arm identification, we focus on the regime where the time horizon $T$ is large in relation to other parameters. This means that we focus on the exponential decay rates of the error/failure probabilities, i.e., the quantity $E^\*  (\nu, d, K)$ in $$\Pr\big[i_{\mathrm{out}}^{\Pi^*}\ne 1\big]\le \mathrm{poly}(d, K)\exp\big( - T\  E^\* (\nu,d,K) \big),$$
>
>     where $\Pi^*$ denotes the optimal best arm identification algorithm, i.e., the one that minimizes the failure probability over $T$ time steps. Note that the polynomial term does not affect $E^\*(\nu,d,K)$ as the above displayed equation can be written as
>   $$\Pr\big[i_{\mathrm{out}}^{\Pi^*}\ne 1\big]\le  \exp\big( - c\  T\  E^\* (\nu,d,K) \big)$$
>
>     for any multiplicative constant $0<c<1$ and for $T$ large enough. Therefore, the polynomial prefactor $\Omega(\frac K d +\log_2 (d))$ can be omitted and only the exponential terms dominate the overall failure probability. In addition, to handle the pre-exponential multipliers or other messy terms in the upper or lower bounds, existing works (e.g., [2, 5-7]) often assume a lower bound on $T$ or simply absorb them into a constant $c$ in the exponent (as in the above displayed equation), whereas we make all terms in the exponent and pre-factors explicit. Hence, it is difficult to compare all bounds quantitatively, except in how the exponential term  $T\   E^\* (\nu,d,K)$ depends on $T$, $d$ and the suboptimality gaps $\\{\Delta_i: i\in[K]\\}$.
>
> Thanks again for your review. Hope these replies resolve your concerns, and any further comments are welcome!
>
> Reference:
>
> [1] Chao Tao, Saúl Blanco, and Yuan Zhou. Best arm identification in linear bandits with linear dimension dependency. In International Conference on Machine Learning, pages 4877–4886. PMLR, 2018.
>
> [2] Alexandra Carpentier and Andrea Locatelli. Tight (lower) bounds for the fixed budget best arm identification bandit problem. In Conference on Learning Theory, pages 590–604. PMLR, 2016.
>
> [3] Emilie Kaufmann, Olivier Cappé, and Aurélien Garivier. On the complexity of best-arm identification in multi-armed bandit models. Journal of
> Machine Learning Research, 17(1):1–42, 2016
>
> [4] Zohar Karnin, Tomer Koren, and Oren Somekh. Almost optimal exploration in multi-armed bandits. In International Conference on Machine Learning, pages 1238–1246. PMLR, 2013.
>
> [5] Ayya Alieva, Ashok Cutkosky, and Abhimanyu Das. Robust pure exploration in linear bandits with limited budget. In International Conference on Machine Learning, pages 187–195. PMLR, 2021.
>
> [6] Julian Katz-Samuels, Lalit Jain, Zohar Karnin, and Kevin Jamieson. An empirical process approach to the union bound: Practical algorithms for combinatorial and linear bandits. Advances in Neural Information Processing Systems, 33, 2020.
>
> [7] Jean-Yves Audibert, Sébastien Bubeck, and Rémi Munos. Best arm identification in multi-armed bandits. In COLT, pages 41–53, 2010.

---

### Official Review · Reviewer_4Xns · 2022-07-11

**Rating:** 5
**Confidence:** 3
**Soundness:** 2 fair
**Presentation:** 2 fair
**Contribution:** 2 fair

**Summary:**

This paper proposes a study of fixed budget BAI with linear bandits. The authors derive a minimax lower bound for linear BAI based on the lower bound provided by Carpentier and Locatelli and propose an algorithm with an upper bound matching the lower bound.

**Questions:**

The following paper may be a relevant study (the authors do not have to cite it, as it is not peer reviewed),

https://arxiv.org/pdf/2202.09036.pdf

**Limitations:**

See the above comments.

**Strengths And Weaknesses:**

The authors' work is based on ideas that seem natural and convincing in the context of BAI study. The theoretical contributions are not surprising and the results may be somewhat obvious given the existing context. However, it is a result that deserves to be adopted in this field. Therefore, I vote to weak accept.

---- After receiving a reply from the author ----

I thank the authors for their replies to my comments. First, "weak accept" in my comment was a typo for "borderline accept". After receiving your reply, I checked the submitted manuscript again, but the scores were still the same.

This study deals with BAI for linear bandits in a fixed budget setting, which has not been well studied. In that sense, it is interesting. However, as a high-level comment, I felt that there is little strong technical novelty. In that sense, I do not think it makes a strong contribution.

In conclusion, while I do not strongly support acceptance, there is no reason to reject it, and there is a novelty in the proposed method. For this reason, I have voted to borderline the scoring, although the score may change after discussion among the reviewers.

---

> ### Author Response · Authors · 2022-07-29
> **Response to Reviewer 4Xns**
>
> We thank the reviewer for the meticulous review and pointing out the relevant paper. This paper proposed a new model of bandit experiments, where the delayed rewards of arm pulls depend on potentially nonstationary contexts. The performance metric studied therein is a linear combination of the stopping time and the simple regret, which is somewhat related to best arm identification. It is interesting to study whether their method, namely deconfounded Thompson sampling, can be extended to linear bandits, especially in the fixed-budget setting. We will discuss this paper more in the revised version of our paper as it is indeed a related work in pure exploration.
>
> As an aside, the esteemed reviewer wrote in her/his review that ''Therefore, I vote to **weak accept**''. However, the score provided is a 5, which corresponds ''borderline accept''. On the other hand, in this conference, ''weak accept''  corresponds to **a numerical score of 6**. Hence, we would greatly appreciate it if the reviewer can up her/his score to a 6 to better reflect her/his sentiments about the paper.
>
> Any further comments are welcome!

---

> > ### Author Response · Authors · 2022-08-02
> > **Author Response (after receiving the reviewer’s reply)**
> >
> > Thanks for your prompt reply! Thanks, in particular, for your response to our query regarding weak/borderline accept. If you have any further comments or questions, we are more than happy to answer them in the author-reviewer discussion period.

---

### Official Review · Reviewer_MHo7 · 2022-07-12

**Rating:** 8
**Confidence:** 3
**Soundness:** 4 excellent
**Presentation:** 4 excellent
**Contribution:** 3 good

**Summary:**

The paper studies the problem of best arm identification in linear arms, with a budget $T$. A key element in their algorithm is the utilization of a G-optimal design to select arms, resulting in a parameter-free algorithm at each steps eliminates non-optimal arms. The authors provide theoretical results for the probability error depending only on the top $d$ arms, where $d$ is the effective dimension of the arm vectors seen as a vector space. The authors compare their algorithm to other previous algorithms and prefer experiments showcasing the empirical and theoretical improvements of their algorithm from the previous ones.



**Questions:**

Would it make sense to add "with budget" to the title of the paper?

In line 306, you write "cannot be improved in an order-wise sense in the exponent in general." Could you please explain what is meant by "order-wise" in this context?

**Limitations:**

The authors do not seem to mention any real-life applications that their work can have. They do mention other contexts where bandits are applied in the real world, but not including a budget setting. I think they would benefit from adding a potential application in industry of their work.

They also do not discuss any negative societal impact of their work, but I think for theory papers this is a bit harder to do.

**Strengths And Weaknesses:**

Strengths

Their algorithm has several improvements from previous algorithms in the literature. In particular, while the idea of partitioning the set of arms into pieces and eliminating certain pieces of the set at each step isn't new, most divide the budget into $\log_2 K$ phases, where $K$ is the number of arms. The author(s)'s algorithm divides the budget into $\log_2d$ phases, where $d$ is the effective dimension of the set of arms as a vector space. Moreover, when compare to Peace in Katz-Samuels, the algorithm does is fully parameter-free, and does not need that suboptimal arms satisfy a certain inequality is not satisfied. If the inequality is not satisfied, the linear bandit instance needs to be rescaled before Peace is run, resulting in a larger bound on the error probability. Furthermore, the exponential term of the error probability is improved by logarithmic factors from Katz-Samuels et al (2020), Ayya et al (2021), and Azizi et all (2021). Finally, it seems their algorithm performs well empirically.

Another important point is that their lower bound is tight and their algorithm is minimax optimal up to multiplicative factors in the exponent.

Weakness:

The authors do not seem to directly address potential applications and limitations of their work. I think they would benefit from including a real world example where linear bandits with a budget is used.

Minor typographical error
Line 182: $\theta^*$ ... change the "$\cdots$" for "$\ldots$"

---

> ### Author Response · Authors · 2022-07-29
> **Response to Reviewer MHo7**
>
> We thank the reviewer for the appreciation of our paper and the constructive comments. We would like to respond to the concerns in the following.
>
> - It is certainly worth discussing more real-life scenarios that our model can capture, especially in the fixed-budget setting. In Appendix E, we conduct an experiment on the Abalone dataset. The age of each abalone is usually hard to determine so it is tempting to predict the age using the other 8 attributes from basic physical measurements, which are cheap and accurate. In this application, our work can be used to adaptively maximize the probability of finding the oldest abalone with a given budget of age measurements. We will discuss more real-life applications of linear bandits in a future version in which we will have one more page.
>
> - We will change the title to "Minimax Optimal Fixed-Budget Best Arm Identification in Linear Bandits" in the final version of this paper. Thanks for your kind suggestion.
>
> - What we mean by "order-wise" in line 306 is the following. If we consider the exponential rate of decay of the failure  probability of the best possible algorithm $\Pi^*$ (i.e., the one that minimizes the failure probability over $T$ time steps), i.e., the quantity $E^\*  (\nu, d, K)$ in
>     $$
>     \Pr\big[i_{\mathrm{out}}^{\Pi^*}\ne 1\big]\le \mathrm{poly}(d, K)\exp\big( - T\  E^\* (\nu,d,K) \big),
>     $$
>
>     then our results imply that there always exists an instance $\nu$ such that $$\underline{c} \cdot(H_{2,\mathrm{lin}}\log d)^{-1}\le E^\* (\nu,d,K)  \le  \overline{c}\cdot (H_{1,\mathrm{lin}}\log d)^{-1}$$
>
>     for some universal constants $0<\underline{c}<\overline{c}<\infty$. Since $H_{1, \mathrm{lin}}\ge H_{2,\mathrm{lin}}$ in general, the bounds are **tight** up to constants. (In fact, it also shows that for the constructed instance in the lower bound, these two hardness quantities are of the same order, i.e., $H_{2, \mathrm{lin}}\le H_{1,\mathrm{lin}} \le c \  H_{2, \mathrm{lin}}$ for a universal constant $c\gt 1$.)
>
> Thanks again for your review. Hope these replies resolve your concerns, and any further comments are welcome!

---

### Official Review · Reviewer_W2wD · 2022-07-12

**Rating:** 7
**Confidence:** 3
**Soundness:** 3 good
**Presentation:** 3 good
**Contribution:** 3 good

**Summary:**

The main contributions:
1. Novel Algorithm for Linear Bandits
2. Upper and Lower performance bounds
3. Numerical Experiments
4. (smaller) comparision and collation of existing methods.

The authors propose a new algorithm that combines various advancements of previous work and add their own novel contribution in the usage of G-optimality. They then proceed to show differences to existing algorithms and to prove lower and upper bounds. The proof for mini-max is argued verbally and more loosely than the rest of the paper. Numerical results then demonstrate the effictiveness on one experimental setup.


**Questions:**

Questions:
1. What do you mean by \delta_1 = \delta_2 in line 104? (I assume that you mean the value is the same of the deltas, is that true for all of them?)
2. It is unclear in the presentation of the paper whether the vectors a(i) are available to the algorithm or not. (The reviewers assumption is that they are).
3. The verbal argument of Mini-max needs to be improved, as it actually shows that it isn't a strict mini-max, rather only an approximate one. The suggestion would be to analyse this better and to show which problem instances lead to what bounds.
4. (smaller suggestion) Also the multiplicative constants of the mini-max bound would be interesting to analyse.
5. Perhaps adding more details on the experimental setup would be helpful too (as opposed to looking up references).

**Limitations:**

Limitations:
1. Mini-Max is perhaps not strictly satisfied and so the core claim of the paper is slightly in question.

Societal Impact:
1. Everything seems to have been addressed.

**Strengths And Weaknesses:**

Strenghts:
1. Strong and Novel (and complex) Algorithm
2. Good mathematical analysis and comparison to other papers
3. Good Upper and Lower Bound Proofs
4. Good Numerical Results.

Weaknesses:
1. Weak verbal "proof" of Mini-max.

---

> ### Author Response · Authors · 2022-07-29
> **Response to Reviewer W2wD**
>
> We thank the reviewer for the valuable comments and suggestions on our paper. Please find our response to the questions below.
>
> 1. We set the gap from the first (best) arm to the second $\Delta_1$ to be the same as the gap from the second arm to the first $\Delta_2=p(1)-p(2)$. This is purely for ease of notation. Although there is no optimality gap for the best arm (i.e., arm $1$), the quantity $\Delta_1$ appears frequently in the theoretical analysis. In particular, it makes the definitions of $H_1=\sum_{1\le i \le K}\Delta_i^{-2}$ and $H_{1,\mathrm{lin}}=\sum_{1\le i \le d}\Delta_i^{-2}$ much more concise as they involve sums from $1$ to $K$ or $d$.
>
> 2. The arm vectors $\\{a(i): i\in[K]\\}$ are available/known to the agent; this is a convention in linear bandits. We will make it clear in the revised form of our paper.
>
> 3. **On the proof of the minimax lower bound**: We believe there may be a misunderstanding regarding the proof of our minimax lower bound which can be found in its entirety in Appendix D in the supplementary material, starting from Line 597.  This is a minimax lower bound in the sense that with respect to the difficult instances, whose construction is clearly presented in the proof, no algorithm can have failure probability smaller than what are prescribed on the right-hand sides of the inequalities in Theorem 3. Our proof is certainly not "verbal" in nature and is delineated in detail in Appendix D.
>
> 4. Multiplicative constants of the minimax bound: In fact, all the multiplicative constants are explicitly stated. Although these constants differ by seemingly large amounts in the upper and lower bounds, we made no attempt to optimize them; rather the spotlight is shone on how the exponents depend on various intrinsic hardness parameters such as $H_{1,\mathrm{lin}}$ and $H_{2,\mathrm{lin}}$.
>
> 5. Due to space limitations, additional implementation details and numerical results (including two other datasets) are postponed to Appendix E. We will try to make the experimental setup more detailed in the main text of a future version in which we will have one more page.
>
> Thanks again for your review. Hope these replies resolve your concerns, and any further comments are welcome!

---

### Meta-Review · Area_Chair_DUgv · 2022-08-27

**Recommendation:** Accept
**Confidence:** Less certain

**Metareview:**

This paper proposed an algorithm and presented a good theoretical evaluation. In particular, the tight lower bound is a nice theoretical contribution.
It is also good that it appears to be descriptively and mathematically sound.

The novelty of the algorithm has been questioned by some reviewers, but even with that contribution, it is shown to have some sufficient advantage.


**Award:**

No

---

### Decision · Program_Chairs · 2022-09-14

Accept